# A Novel and Extensible Remote Sensing Collaboration Platform: Architecture Design and Prototype Implementation

Wenqi Gao [1,2], Ninghua Chen [1,2,*], Jianyu Chen [3,4], Bowen Gao [1,2], Yaochen Xu [1,2], Xuhua Weng [1,2] and Xinhao Jiang [1,2]

1   Key Laboratory of Geoscience Big Data and Deep Resource of Zhejiang Province, School of Earth Science, Zhejiang University, Hangzhou 310058, China; gaowenqi@zju.edu.cn (W.G.); gaobw@zju.edu.cn (B.G.); 11838005@zju.edu.cn (Y.X.); xuhua.weng@zju.edu.cn (X.W.); xinhaojiang@zju.edu.cn (X.J.)
2   School of Earth Sciences, Zhejiang University, Hangzhou 310058, China
3   State Key Laboratory of Satellite Ocean Environment Dynamics, Second Institute of Oceanography, Ministry of Natural Resources, Hangzhou 310012, China; chenjianyu@sio.org.cn
4   School of Oceanography, Shanghai Jiao Tong University, Shanghai 200030, China
*   Correspondence: geo316@zju.edu.cn

**Abstract:** Geospatial data, especially remote sensing (RS) data, are of significant importance for public services and production activities. Expertise is critical in processing raw data, generating geospatial information, and acquiring domain knowledge and other remote sensing applications. However, existing geospatial service platforms are more oriented towards the professional users in the implementation process and final application. Building appropriate geographic applications for non-professionals remains a challenge. In this study, a geospatial data service architecture is designed that links desktop geographic information system (GIS) software and cloud-based platforms to construct an efficient user collaboration platform. Based on the scalability of the platform, four web apps with different themes are developed. Data in the fields of ecology, oceanography, and geology are uploaded to the platform by the users. In this pilot phase, the gap between non-specialized users and experts is successfully bridged, demonstrating the platform's powerful interactivity and visualization. The paper finally evaluates the capability of building spatial data infrastructures (SDI) based on GeoNode and discusses the current limitations. The support for three-dimensional data, the improvement of metadata creation and management, and the fostering of an open geo-community are the next steps.

**Keywords:** geospatial data services; remote sensing; user collaboration; customized web app; geo-community

## 1. Introduction

Over the past few decades, a significant number of Earth-monitoring satellites have been deployed by space agencies [1]. RS data, along with other geospatial data, have seen exponential growth. These data not only hold tremendous significance for scientific research [2], but also revolutionize our lives [3,4]. RS is widely used in scientific research in many fields, including economics, ecology, transportation, and archeology. Among them, in the field of Earth science, Li et al. [5] used RS images and digital elevation models to discover the ancient lake and reconstruct the ancient climate in the Gobi. Dervisoglu [6] investigated the changes in the long- and short-term water surface area in Turkey. Wang et al. [7] utilized satellite images to study the geological features and evolution of yardangs. Tong et al. [8] integrated RS and numerical modeling to quantify the warming trends of lakes. These efforts are dedicated to understanding the Earth's historical evolution and envisioning its future, analyzing global changes to create a sustainable Earth environment. There are also many application cases in production activities and public services. Karimli [9] used Sentinel-2 raster data and SRTM DEM data to calculate vegetation and

soil indices to estimate winter wheat yield. Su et al. [10] used fishery data and satellite images to model the spatial distribution of swordfish to assist fishery management and conservation. McCormack [11] applied Sentinel-1 to reconstruct flood-level timeseries at seasonal wetlands in Ireland and demonstrated its potential as a viable tool for the remote monitoring of ecologically significant wetlands. Zhang [12] analyzed the economic connection network structures within urban agglomerations in the new western land–sea corridor in Western China based on nighttime light RS.

For RS applications, the process from raw data to information and ultimate knowledge necessitates needs domain expert knowledge. RS data processing comprises atmospheric correction, georeferencing, information extraction, etc. [13]. These processes highly require specialized knowledge, such as knowledge of radiative transmission theory, coordinate systems, photogrammetry, and spectral signatures [14]. The professional processing challenges non-specialized users, who have demands for geospatial data services due to the need for interdisciplinary research, management, decision-making, or even simple interests. Nevertheless, non-specialized users need to undergo training to become proficient in using RS data [15].

In the field of geographic information, significant technological advancements have been achieved in the past three decades [16]. The advent of the web-based GIS provides a powerful online platform to support the development of geospatial information [17]. Geoportals are one of the main components of spatial data infrastructures (SDIs). They serve as robust internet-based platforms designed for the discovery and retrieval of geospatial information. Additionally, these portals facilitate various geographic services, including sharing, displaying, analyzing, and publishing geographic information online [18]. In recent years, geospatial services have experienced rapid development. Friis-Christensen et al. [19] introduced the concept of distributed geographic information processing and explored different architectural patterns for collaborative geoprocessing services. Geosquare provides a cloud-based framework for building, executing, and sharing collaborative models [20]. Ma et al. [21] designed a customizable process expression model and proposed a corresponding process customization method for collaborative geographic analysis. These efforts fully discuss the collaborative patterns for geospatial services and explore novel methods for geospatial applications. However, existing geospatial service platforms are more oriented towards users with specialized expertise, such as GIS developers and spatial analysts, in both the implementation process and final application. Building appropriate geographic applications for non-professionals remains a challenge.

To make it easier for the public to use RS data, meet the diverse usage requirements of end-users across various levels, and enhance collaboration between experts and non-experts, this study proposes a geospatial data service (GDS) architecture and develops a platform based on GeoNode [22]. The platform mainly consists of four parts: the geospatial content management system (Geo-CMS) module, the interactive mapping module, the data processing and analysis workflow, and the geo-community. This work allows users to integrate web apps into the platform. The platform enables efficient data management, providing experts with a space to apply professional knowledge and offering a pathway for general users to access RS data services.

## 2. Methods

### 2.1. The Architecture for the GDS Platform

The RS data service collaboration platform functions as an internet-based geoportal and provides universal entrances for various RS applications in geology, ecology, oceanography, etc. Both experts and general users can obtain production tools and customizable data services. The platform connects various user groups, eliminating barriers to cross-disciplinary collaboration and promoting the sharing of geospatial data. Ultimately, it aims at creating an inclusive geospatial community that welcomes participation from individuals of all backgrounds.

The overall architecture of the platform is presented in Figure 1. The scheme underlines the interaction between potential end-users and external applications. A collaborative platform of users, web apps, and software for RS data applications is depicted as a whole. The scheme appropriately integrates desktop GIS data processing software (QGIS 3.22 and ArcGIS 10.8) and a geospatial analysis cloud platform (Google Earth Engine, GEE), establishing a flexible and extensible data processing and analysis workflow.

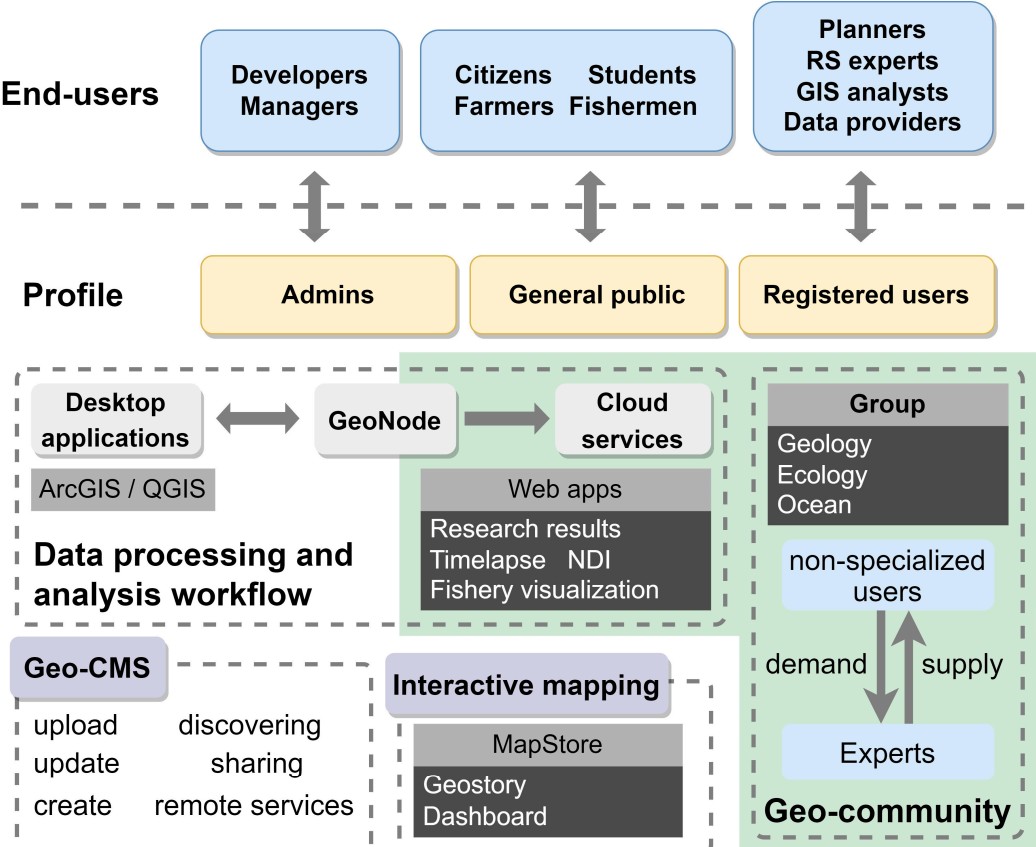

**Figure 1.** Overall architecture of the GDS and interaction diagram. Components identified with the green dashed line were proposed and developed based on GeoNode.

The main features of the GDS include data management, interactive mapping, and user re-creation. Data management involves essential geoportal functions (e.g., uploading, sharing, discovering, etc.). Interactive mapping and user re-creation empower users to create immersive content (e.g., Dashboard; GeoStory). Three extended web apps offer great data analysis capabilities. In addition, we create distinctly themed forums to satisfy the demands of non-specialized users. All of these features are seamlessly integrated into a unified graphical user interface (GUI) (Figure 2), offering users a one-stop solution to fulfill their requirements.

*2.2. Interactive Solution with Leading Web Technology*

The GDS platform construction plan presented in this paper is characterized by its simplicity in implementation and inclusivity. We adopt a free and open-source software (FOSS) technological stack combining several software packages to achieve this goal. After evaluating factors such as user base, operational and maintenance status, and the quality of documentation, we conducted a comprehensive survey of each software or package and identified the appropriate ones. Figure 3 illustrates the technologies and tools employed in this paper.

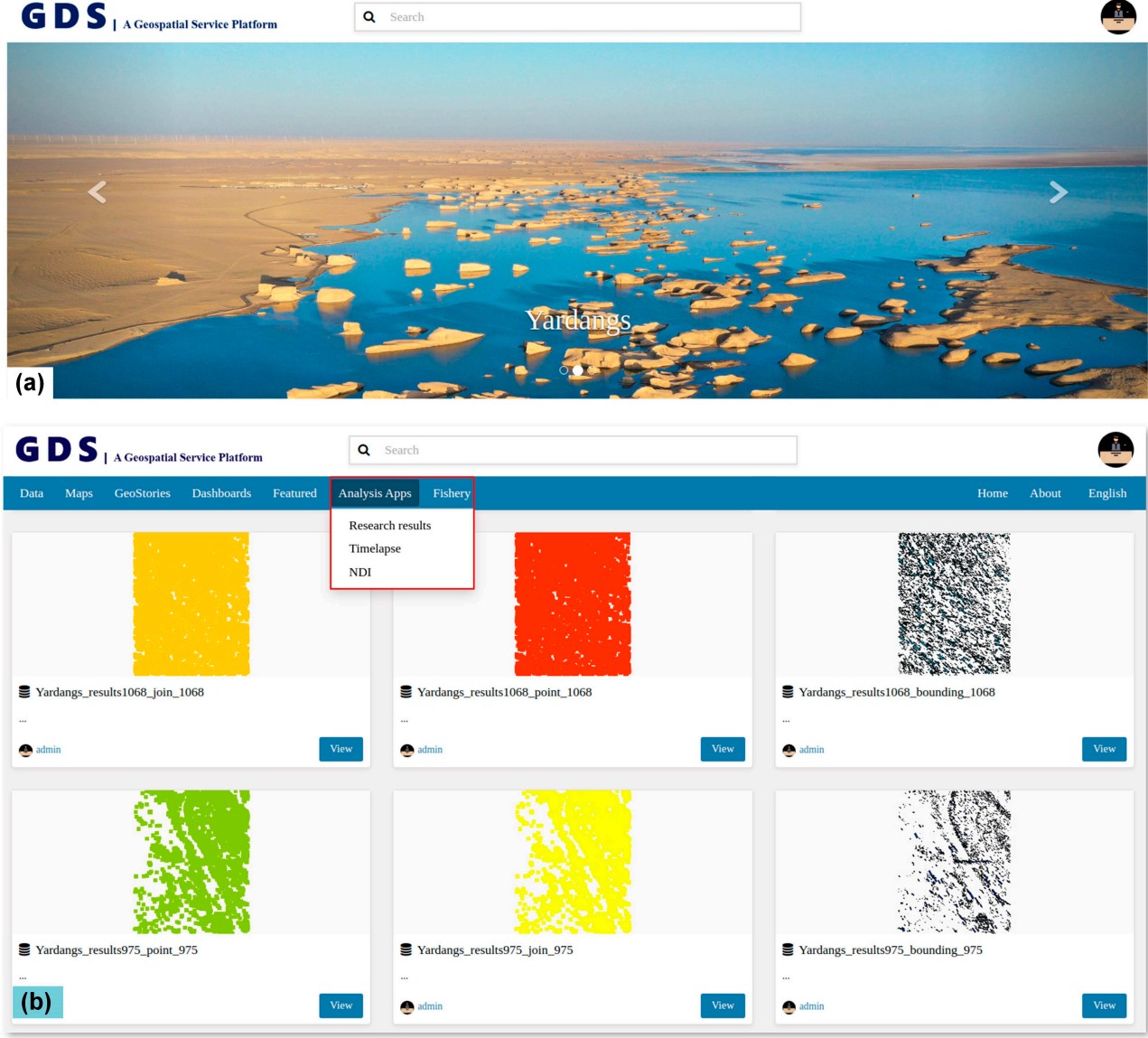

**Figure 2.** The GUI of the GDS platform. (**a**) A slideshow-style homepage interface, where (**b**) the upper part consists of a menu bar and the entrance of the web apps, represented by a red rectangle. The lower part displays a goods shelf for the platform's data.

Among the abundant open-source geospatial software options, GeoNode stands out as a mature and dependable choice. Developed by the Python web framework Django, GeoNode integrates many stable and robust open-source projects instead of starting from scratch [22]. GeoNode's built-in tools provide a wealth of user-friendly features, including map creation, data visualization, and an advanced GUI for browsing diverse data. Its distinctive social attributes play a crucial role in forum creation and community development. To meet the requirements of different language-speaking users, a language switch option is incorporated. Presently, it supports Chinese and English. With further development of the platform, multilingual support will be better achieved.

The presentation layer is built on the top of AngularJS and Bootstrap, as well as MapStore2 for the mapping applications. Geoserver, GeoWebCache, and Pycsw together constitute the application layer. Nginx and Apache Tomcat are employed to handle HTTP requests. In the data layer, PostgreSQL, supplemented with the PostGIS extension, serves as the geospatial database management server for storing vector datasets. Raster datasets are stored in the file system, and Elasticsearch facilitates knowledge discovery.

In addition to the GeoNode-based part, we utilize the remarkable scalability of the Django web framework to develop customized web apps. Django supports the development of dynamic websites, applications, and services. It uses MVC pattern and divides an application into three layers: Model, View, and Controller. Geemap is designed for students and researchers who utilize the Python ecosystem of diverse libraries and tools to explore Google Earth Engine [23]. They jointly compensate for the platform's RS data computational limitations. Efficient data management capabilities, outstanding visualization performance, integration with advanced data processing tools, and the link to the cloud computing platform make GDS a reliable option to process and analyze data with.

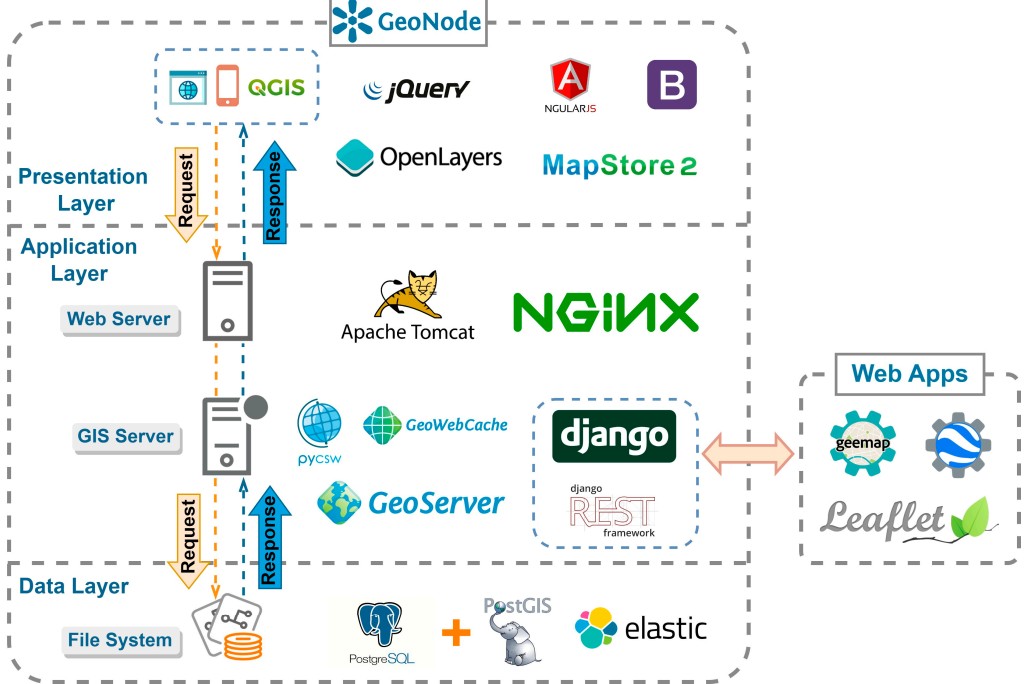

**Figure 3.** The GDS platform's technology solution. The left part depicts the core platform built on the GeoNode framework, while the right part represents extensible web apps.

*2.3. The Collaboration between General Users and Domain Experts*

As depicted in Figure 1, the GDS platform serves a variety of end-users, such as managers, citizens, students, farmers, RS experts, GIS analysts, etc. To increase the accessibility of the data and build a geospatial community that everyone can participate in, we choose the simplified user management mechanism. Users are categorized as Administrators, General public (unregistered users), and Registered users. Groups are another internal way to organize users, where Administrators can create forums for different domain users (Figure 1).

It is vital for the platform's progress to determine whether or not users can easily locate the data that meet their specific needs. However, the existing data might not meet all application scenarios, and non-specialized users often struggle to locate satisfactory data. In these situations, leveraging expert knowledge to solve users' challenges is a sensible approach. Utilizing the social attribute of GeoNode, a collaborative mode (Figure 4) is implemented, enabling non-expert users to submit their requirements and experts to respond to them. Users can accurately post their data needs in the relevant group. Experts then review the requirements, engaging in further discussion with users to confirm details or using existing knowledge and data to align with users' requests.

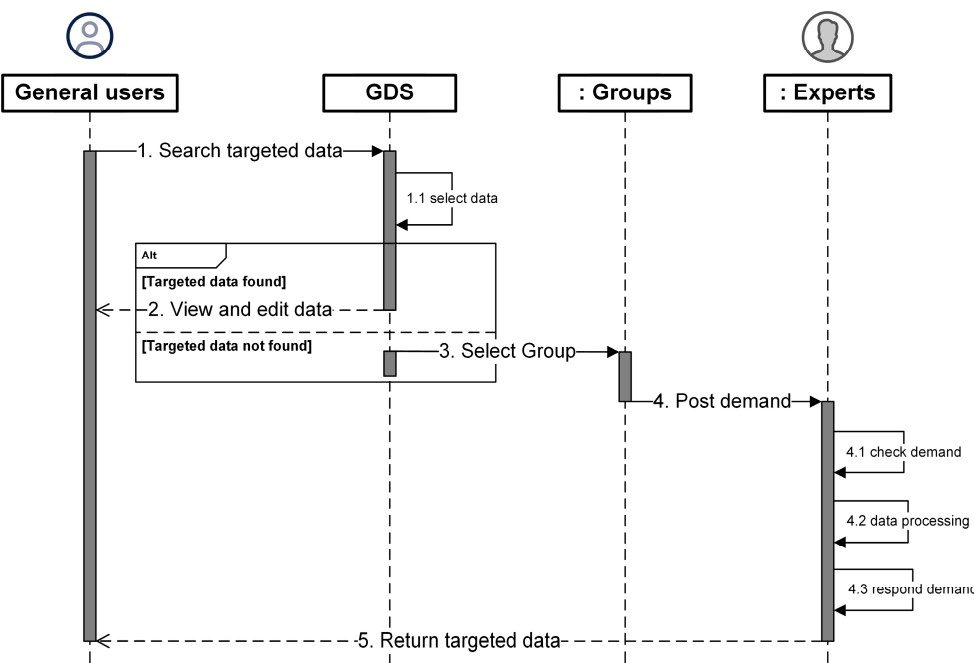

**Figure 4.** Data collaboration sequence diagram for non-specialized users and experts. From left to right, the process for general users to find the required data within the platform is explained, as well as the interaction with experts.

This collaborative mode acts as a bridge between regular users and experts. Additionally, the GDS can serve as a data source connected to desktop GIS software, such as ArcGIS 10.8 and QGIS 3.22. Experts can access existing WMS and WFS map services on the GDS, and employ processing tools to perform advanced data processing and analysis. Users can also obtain results from experts in this way. During the prototype phase that was implemented for the platform, this collaboration takes the form of volunteered geographical information (VGI). To ensure sustainable development, the introduction of incentive policies is considered to encourage more users and experts to participate [24].

## 3. Datasets and Documents

RS data and other geospatial data are core components of the GDS platform. During the development phase of the GDS platform, users uploaded data from three research domains: ecology, geology, and oceanography. These data involve both foundational datasets and research outcomes. In this process, we took strict quality control measures supervised by experts with specialized knowledge to ensure the accuracy and dependability of these data for users. The GDS platform's inherent dynamic and collaborative nature allows end-users to continuously improve and refine the collected datasets.

Figure 5a shows statistical information regarding the available data on the platform. As of August 2023, the platform has stored 983 geospatial datasets in total as well as meta-data. These datasets come from various sources, including online databases (e.g., National Cryosphere Desert Data Center, China; Earth Science Data System, America, etc.), re-creation from experts and users, as well as the collection and arrangement of existing research results.

In general, ecological data make up 46% of the total (Figure 5c), primarily consisting of global mangrove distribution, aboveground biomass, and canopy (80%) date, along with DEM and topographical data (13%), and various vector data (7%). Geological data constitute 36% of the total (Figure 5d), mainly comprising yardangs and related research data (56%), published geological theses (18%), and land cover vector data (7%). Oceanic data account for 18% of the total (Figure 5b), with the majority being chlorophyll data

(58%), followed by published ocean research (20%) and fisheries data (17%), along with other ocean-related datasets.

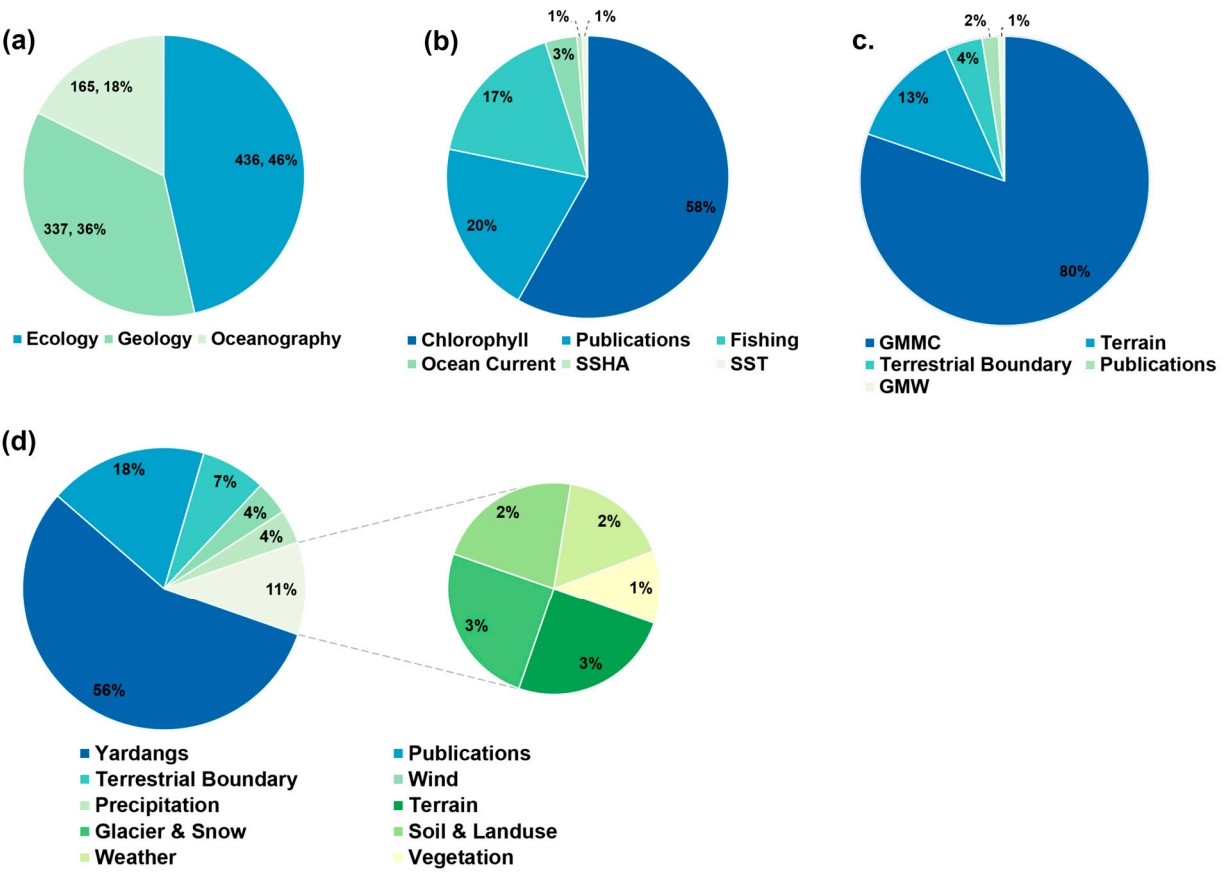

**Figure 5.** Existing data statistics chart on the GDS. (**a**) Number and proportion of ecological, geological, and oceanic data; (**b**) detailed statistics on oceanic data; (**c**) detailed statistics on ecological data; (**d**) detailed statistics on geological data.

## 4. Use Cases

### 4.1. Geology Case

Using the data introduced in Section 3, this section presents the collaboration between non-specialized users and experts of the GDS platform, with a focus on its significant role in geological research and the understanding of geology-related knowledge.

Yardang was a geological term coined by Hedin in 1903 during his expedition to northwestern China's Lop Nur region. It refers to the streamlined ridge formed through wind erosion within lacustrine deposits [7]. Researchers have qualitatively and quantitatively studied yardangs' morphological characteristics. Numerous controlling factors influence the external morphology and development process of yardangs, including lithology, tectonics, wind speed, precipitation, etc. [25]. These findings and associated datasets have been collected and uploaded to the platform. Users can view the yardangs datasets (with varying data access levels depending on user permissions), use existing data for re-creation, and publish their own results. For the users who are unfamiliar with yardangs and struggle to make the best use of these data, we propose a collaboration mode within the GDS platform that bridges the gap between general users and experts.

The geological forum, named Geology, is one of the three existing forums on the platform, open to all registered users. Here, users can freely express their thoughts and needs and communicate via the e-mail they used for registration (Figure 6a). After receiving inquiries or requests from general users, experts offer guidance or suggestions to the users, or they generate the required data, depending on specific requirements.

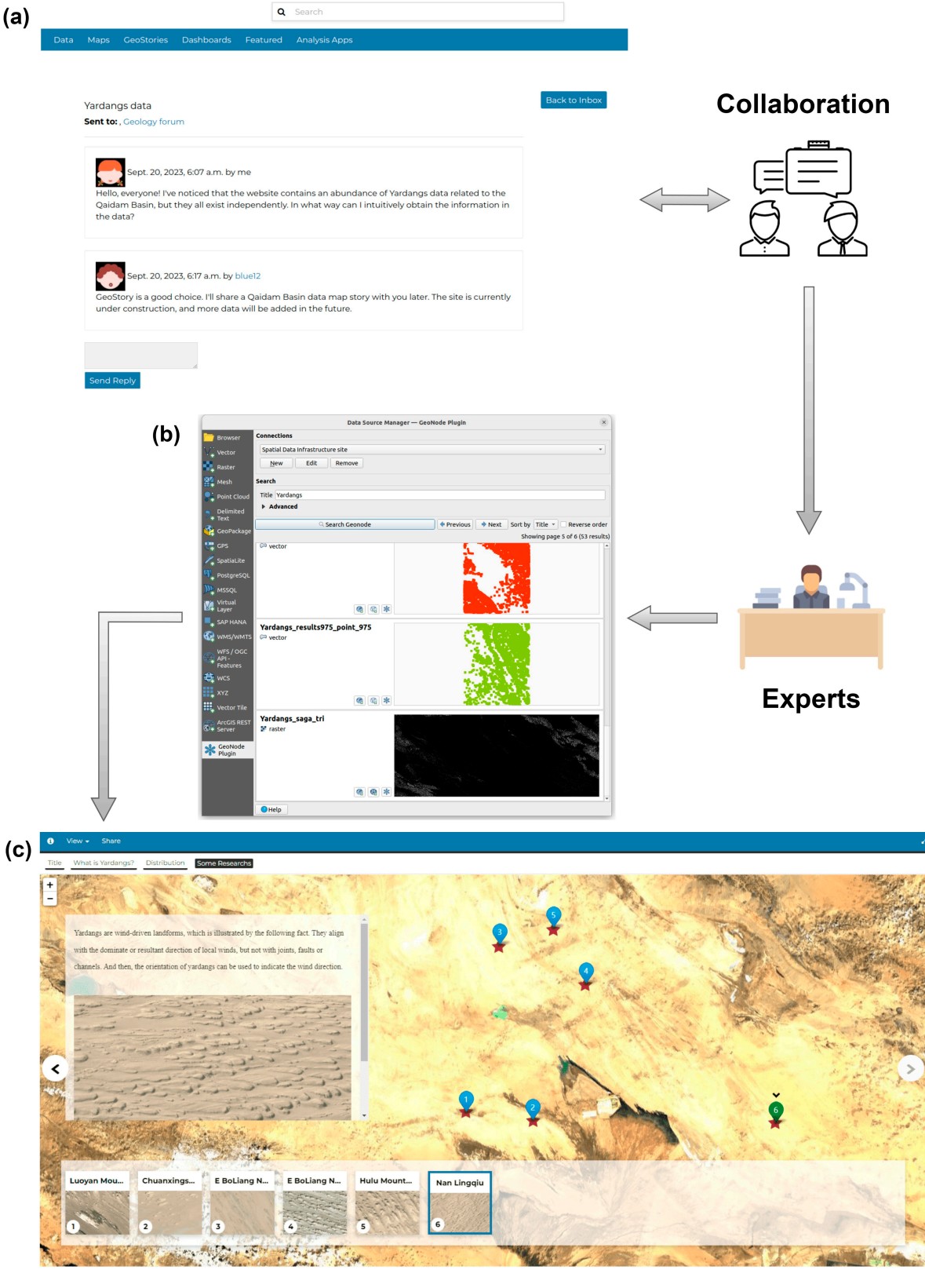

**Figure 6.** User collaboration case diagram. (**a**) Communication diagram featuring requests from general users and expert responses. (**b**) screenshot of the connection between QGIS and the GDS platform; this figure displays the data discovered on the GDS via a keyword search for yardangs (**c**) Yardangs-themed GeoStory; this picture showcases the yardangs landscape in six anticlinal areas (Hulushan, Luoyanshan, Chuanxingshan, etc.) through the GeoCarousel.

Although the platform already boasts many yardangs datasets for the Qaidam Basin, the possibility of it lacking the required data exists. Using the QGIS, we can generate a shapefile representing the distribution of six typical anticlinal areas in the Qaidam Basin, such as Hulushan, Luoyanshan, Chuanxingshan, etc. By linking the QGIS to the site (Figure 6b), a map that illustrates the distribution of these six anticlinal regions is created. GeoStory provides a novel way for users to interact with these maps compared with traditional ones. By integrating maps, text, and multimedia data, this procedure represents an effective visualization method that appeals to user interest and involvement [26,27]. In this case, the GeoStory with the theme of "Yardangs" uses a distribution map of typical anticlinal regions in the Qaidam Basin, along with additional datasets of factors such as wind speed, precipitation, and multimedia documents (Figure 6c). Subsequently, the expert posts it on the platform for the user to view. Specifically, the text section helps users gain a deeper understanding of the objects they are interested in and allows them to add hyperlinks to any content they want users to explore. GeoCarousel provides users with an immersive spatial experience. Multimedia content, such as images and videos, offers additional perspectives on yardangs. Additionally, creators can configure corresponding visibility permissions for sensitive data.

### 4.2. Ecology Case

Spectral indices, derived from multispectral remote sensing products, are extensively used to monitor Earth system dynamics (e.g., water bodies, fire regimes, vegetation dynamics, etc.) [28]. Surface materials have unique spectral signatures due to their physical properties and interactions with electromagnetic radiation. Environmental factors can influence these interactions, and spectral features provide valuable insights into various surface processes, such as urban expansion [29], vegetation growth [30], fire severity [31], etc. To facilitate the computation and analysis of spectral indices, we developed a web app named normalized difference index (NDI) calculation and seamlessly integrated it into the platform (Figure 7). In this app, users can select a specific spectral index, specify the timespan, adjust the playback speed, and control other parameters to calculate the specific index.

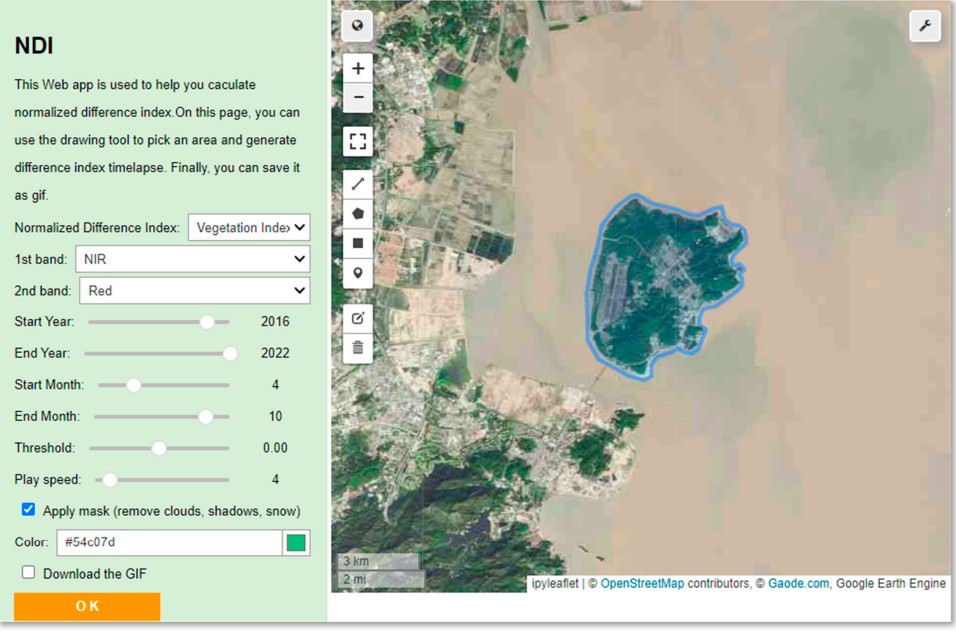

**Figure 7.** The "NDI" web application's GUI. On the left side, users can select spectral indices and other conditions, while the right side displays the resulting images on the map widget. The blue rectangle is the area of interest drawn by the user.

Spectral indices are primarily applied to the terrestrial biosphere, particularly focusing on vegetation monitoring via the vegetation index (VI). The most well-known and frequently used VI is the normalized difference vegetation Index (NDVI), which is calculated as the normalized difference between near-infrared (NIR) and red reflectance. It is commonly used to evaluate vegetation greenness over spatial and temporal dimensions [28]. The mangrove forest ecosystem is highly susceptible to anthropogenic disturbance and climate change, making it one of the most vulnerable ecosystems on Earth [32]. Analyzing the inter-annual changes of mangroves helps decrease uncertainty in quantitative studies. We applied this app to calculate the NDVI of Qi'ao Island (Figure 8) and briefly analyze mangrove growth.

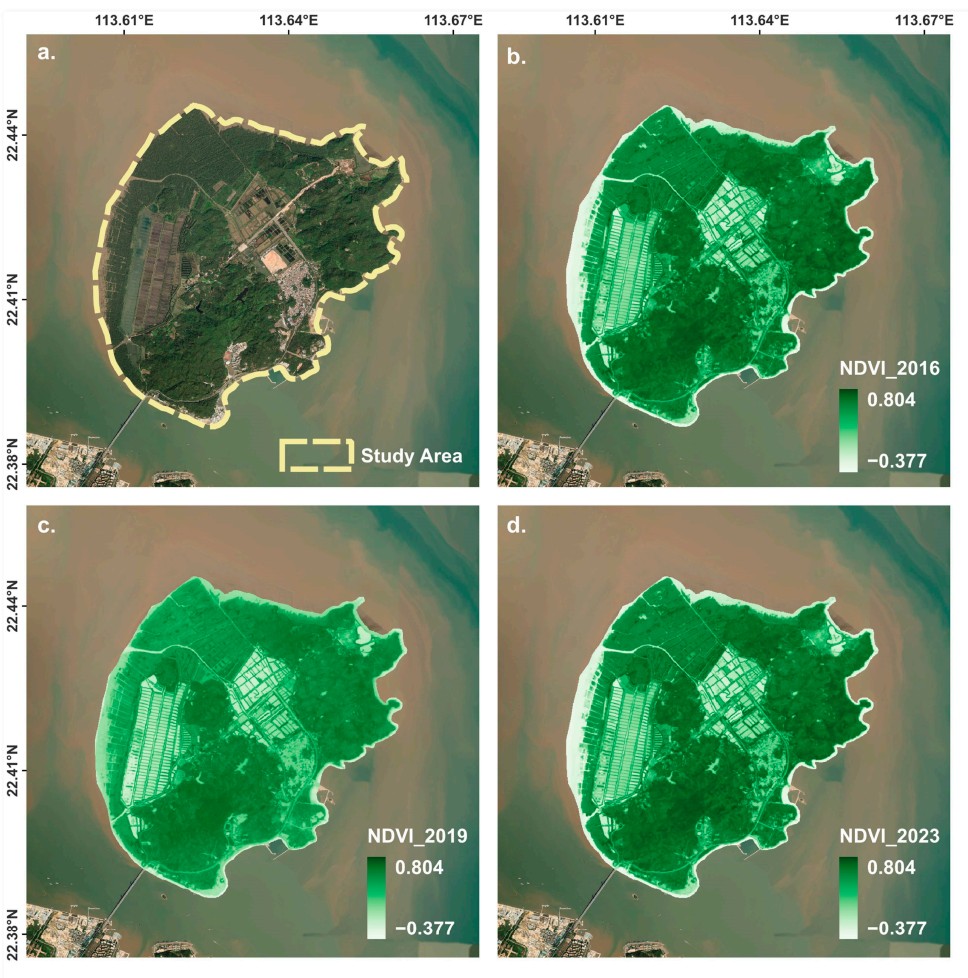

**Figure 8.** The Qi'ao Island and its NDVI across different years. (**a**) The location of Qi'ao Island; (**b–d**) representations of the NDVI calculation results for the years 2016, 2019, and 2023, respectively.

The dashboard, integrated within GeoNode, offers users a space to incorporate various widgets, such as charts, maps, tables, texts, and counters, allowing users to establish connections between these elements. These connections introduce a dynamic and interrelated way of displaying information. We incorporate the base map of Qi'ao Island, the NDVI calculation results, and a few days' NDVI data into a dashboard using different widgets (Figure 9). Overall, the vegetation area of Qi'ao Island gradually increased between 2016 and 2023. The mangrove growth area along the intertidal zone expanded significantly. Some interior areas of the island experienced a reduction in vegetation, and are covered by roads and buildings to accommodate the need for more living space.

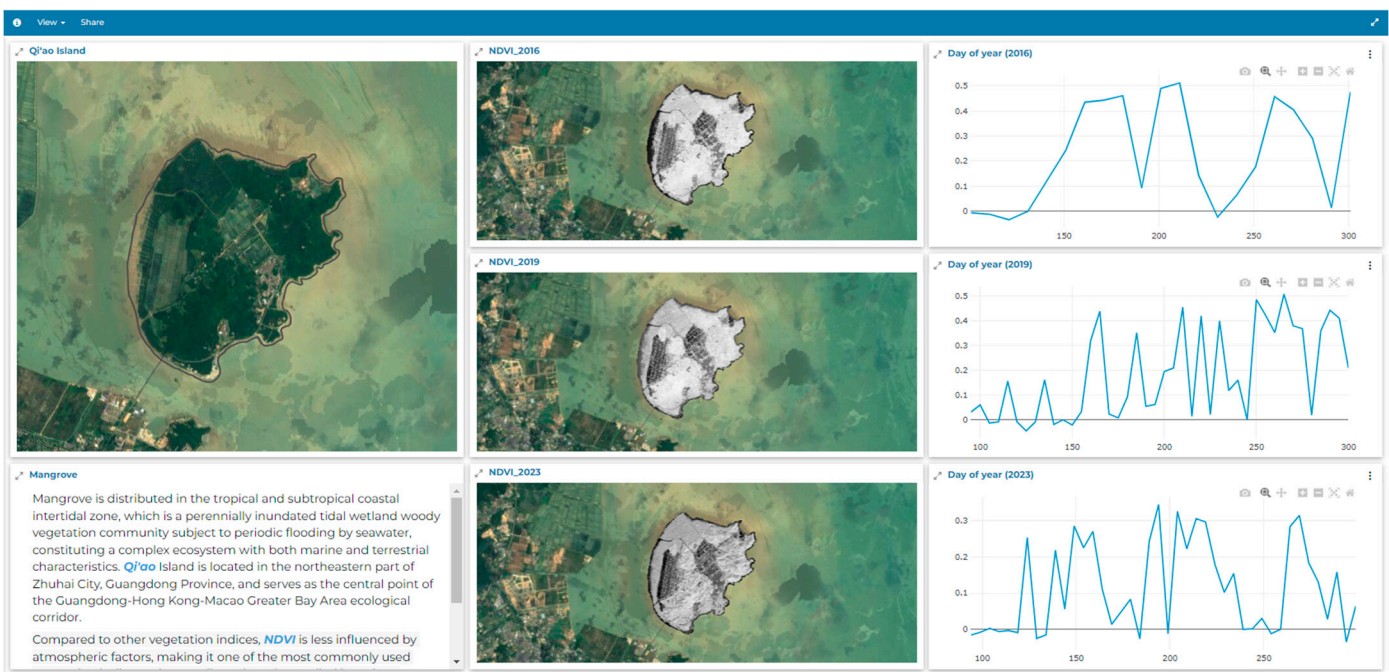

**Figure 9.** A dashboard with the theme of Qi'ao Island's vegetation. It includes three types of widgets: a map, text, and chart.

### 4.3. Oceanography Case

The sustainable exploitation and utilization of the ocean are closely related to human survival and development. Satellite remote sensing data used in oceanographic work are primarily on sea surface temperature (SST), ocean color, and ocean altimetry [33]. Analyses of RS data and fisheries data within a GIS environment have facilitated the elucidation of fundamental relationships between marine biota and their oceanic environment [34].

In our earlier work, the processing and visualization of fisheries data involved several steps (Figure 10). Each step relied on different software and tools. Firstly, we downloaded the raw data from the FTP (file transfer protocol) server to the local computer, a process dependent on software such as FileZilla. These data are of various types such as SST and sea surface height anomalies (SSHA). Next, the Python GDAL (Geospatial Data Abstraction Library) was employed to process the downloaded data. This step involves reading the file header and saving the data body as a GeoTiff file. Finally, we conducted visualization and analysis using GIS software. These complex processes are time-consuming and require users to possess a variety of skills.

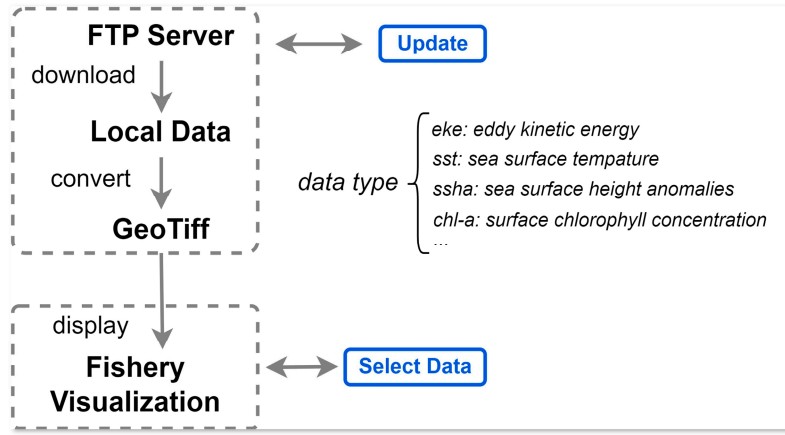

**Figure 10.** Fishery data processing steps and the main included data types.

With GeoNode's extensibility, we simplify the traditional fisheries data processing steps and develop the Fishery Visualization application (Figure 11). The download and convert steps are centrally processed on the back-end services. In addition, we utilize Leaflet to display processed GeoTiff files. Users simply click the 'Update' button to sync data from the FTP server to their local environment. This reduces the time used to process the data and convert data between different formats, giving researchers more time to focus on their investigations. Users can click the 'Select Data' button to choose the files they want to view and assign a color gradient to them. Clicking on the layer allows users to inspect the pixel value. This workflow, which used to require three separate software and tools, can now be accomplished on a single webpage. In the future, more applications will contribute to the sustainable development and utilization of the ocean.

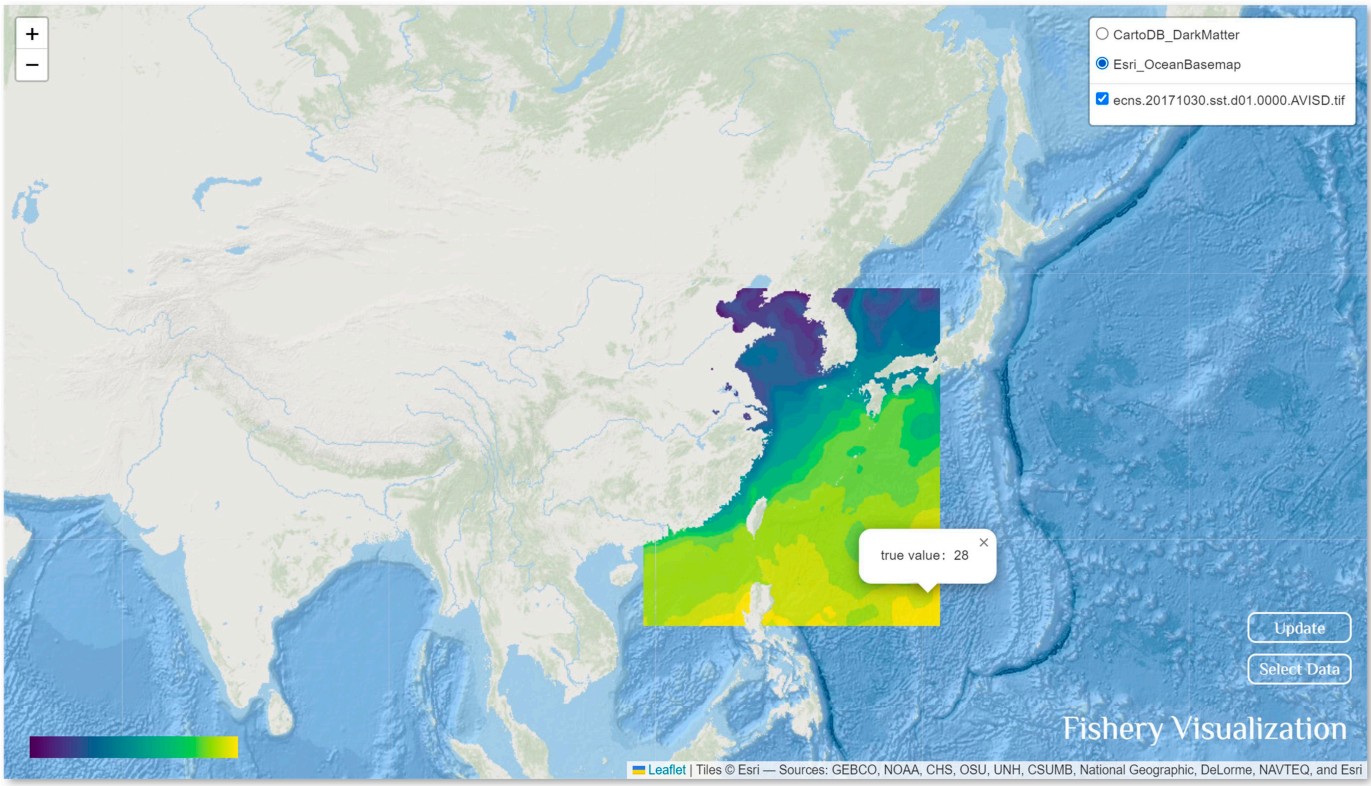

**Figure 11.** The "Fishery Visualization" web application's GUI. This picture contains a loaded sea surface temperature layer.

## 5. Discussion

### 5.1. Capabilities Analysis for Building GeoNode-Based SDI

Enabling everyone to obtain convenient geospatial services has always been the vision of geographic information development. In recent years, technological advancements, especially the growth of mobile internet, have given rise to more abundant application scenarios for geospatial services, which are therefore different from the past where SDI was mostly developed and operated by governments and organizations. More and more small-scale teams (such as labs and interest groups) or individuals are seeking to establish their own lightweight SDI.

From a software engineering point of view, this study adopts an architecture based on GeoNode, avoiding many technical issues commonly encountered in traditional WebGIS development. For those without development experience, creating a WebGIS platform is almost an impossible task. It requires learning substantial amounts of web development and GIS knowledge without prior experience. Firstly, HTML, CSS, and JavaScript form the foundation of web development, playing crucial roles in building web applications. Secondly, it is important to master server-side development skills and become familiar

with popular front–end frameworks (e.g., Node.js, Vue, React, etc.). Finally, to grasp the WebGIS development process and be able to independently complete a small project, it is necessary to learn how to use JavaScript libraries such as OpenLayers, Cesium, and more. This process is time-consuming and poses a significant challenge for individuals without prior experience. GeoNode supports geoportal development by consolidating mature open-source software projects in an easy-to-use interface. Non-specialized users can easily share data and create interactive maps. This approach streamlines the entire process and reduces the development timeline, making it a suitable tool for individuals or small-scale teams. There are existing cases of using GeoNode to build SDI (Table 1). These cases strongly demonstrate GeoNode's adaptability in building geoportals across different spatial scales and application scenarios.

**Table 1.** Details of the spatial data infrastructure developed based on GeoNode (as of November 2023).

| Name | Stakeholder or Coordinator | Scope (International, National, Local) | Website |
|---|---|---|---|
| GeoINTA | National Institute of Agricultural Technology of Argentine | National | https://geo-backend.inta.gob.ar/#/ (accessed on 6 December 2023) |
| GeoMOP | Public Works Ministry of Argentine | National | https://geoportal.obraspublicas.gob.ar/ (accessed on 6 December 2023) |
| Geoportal 3F | Buenos Aires province, Argentine | Local | https://geoportal.tresdefebrero.gob.ar/ (accessed on 6 December 2023) |
| Geoportal Lujan de Cuyo | Mendoza province, Argentine | Local | https://geoportal.lujandecuyo.gob.ar/ (accessed on 6 December 2023) |
| DECAT | Cyprus | National | https://decatastrophize.eu/ (accessed on 6 December 2023) |
| Ocean Observatory | Mauritius | National | https://gococeanobservatory.govmu.org/ (accessed on 6 December 2023) |
| CEPAL | UN—ECLAC | International | https://geoportal.cepal.org/ (accessed on 6 December 2023) |
| THAL CHOR | Greece | National | https://thalchor-2.ypen.gov.gr/ (accessed on 6 December 2023) |

From a platform type of view, the majority of applications listed in Table 1 are static and serve solely as providers of geospatial services. Geo INTA, Geoportal 3F, Geoportal Lujan de Cuyo, Ocean Observatory, and THAL CHOR merely utilize the native functions of GeoNode. CEPAL provides many external links, including NASA and satellite data sources, etc. GeoMOP, led by the Argentinian government, is designed to support public utilities. DECAT was developed towards the better protection of citizens against disaster risks in Europe. Both GeoMOP and DECAT are complex and costly. However, the GDS platform promotes collaboration between general users and experts. Combining Django with Google Earth Engine compensates for the GDS platform's deficiencies in RS data computation and analysis capabilities. Presently, we develop the research results of the visualization application (Figure 12), the temporal landscape change analysis application, and the land cover normalization difference index calculation application. They are seamlessly integrated into the platform. These attempts and experiences also contribute positively to the design and development of multi-scenario geospatial data application service platforms. In the future, the platform will incorporate an increasing number of web applications. This development mode is dynamic and sustainable.

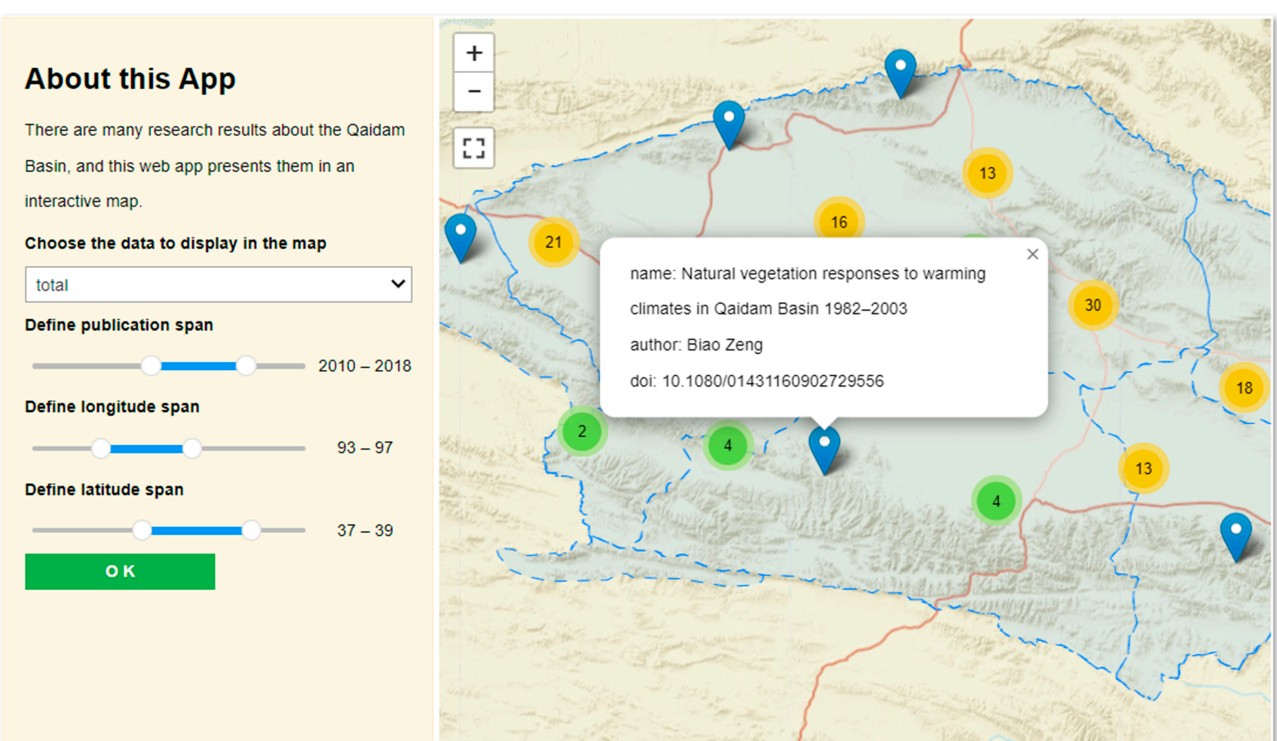

**Figure 12.** The "Research results" web application's GUI. On the left side, users can configure search conditions, while the right side displays the query results on the map widget. The popMarker displays attribute information for the user's query point, including existing research result name, author, and DOI.

### 5.2. Limitations and Future Improvements

The platform now supports various task scenarios, but some limitations still impede its full potential for geospatial data management and comprehensive analysis. To address this challenge and provide better services for users, we aim to build a robust and sustainable geospatial data service platform.

Firstly, the current platform only supports the uploading and visualization of two-dimensional geospatial data. Support for three-dimensional data is further needed. Three-dimensional data with enriched spatial information provide a more comprehensive and realistic perspective for scientific research, planning, and visualization. For instance, LiDAR-scanned point cloud data can be used to create high-precision terrain and building models. Underground geological data describe subterranean structures and strata. Meteorological and climate model data simulate the atmosphere and ocean. Urban 3D planning models are used for urban design and development. The development of digital twins and smart cities has made the demand for three-dimensional data more urgent.

Secondly, the creation and management of metadata should be more refined. Incorrect metadata can disrupt users' normal usage and ultimately lead to significant chaos for the platform. There are two main difficulties in metadata creation and management [35]. The published geospatial data cannot obtain relevant metadata, and the metadata should be kept updated when the data are updated. Therefore, achieving the automatic creation and synchronized updating of metadata on the GDS platform is at the forefront of the next work.

Thirdly, experts play an important role in the operation and development of the collaborative platform. To ensure that there are a sufficient number of experts to address user needs, we will design a clear expert recruitment plan. We will promote this extensively within the field of Earth science, inviting professionals interested in the platform's objectives to participate. We will develop an incentive mechanism to attract expert involvement, and categorize users' needs into regular and special requirements. Resolving regular re-

quirements earns points, and the points correspond to different rewards. For users' special requirements, experts can communicate with users to determine the appropriate incentive method. We will establish an expert community as a space for interaction and communication among professionals, fostering a spirit of collaboration. Furthermore, we will improve the management of experts' information, ensuring that all experts provide support in their experienced and interested fields, and avoiding excessive workload allocation. Establishing an effective communication channel enables experts to communicate directly with the platform's managers to share concerns and suggestions.

Lastly, GIS service providers usually involve users as passive recipients of data and services, resulting in a one-way process. By categorizing users into groups based on their interests, we enhance a sense of belonging and offer a conducive environment for collaborative problem solving. General users shift from solely acquiring data to communicating and collaborating with experts to collectively produce data. This transformation turns the platform into a geospatial community in which content is user-generated. Our efforts are beginning to bear fruit. However, data generation and creation require both general users' and experts' active participation. Confronted with growing ecological challenges and climate change, we will take further actions to promote data sharing in scientific research.

## 6. Conclusions

Geospatial data, especially RS data, hold significant importance for public services and production activities. The steps of data processing, information extraction, and knowledge discovery heavily rely on specialists, making it challenging for non-specialized users to fully use. However, existing geospatial service platforms are more oriented towards professional users in terms of implementation processes and final applications. This study focuses on designing a geospatial data service architecture that links desktop GIS software and cloud-based platforms, constructing an efficient user collaboration platform. The objective is to promote collaboration between experts and the general public, and provide geospatial data services across various domains. This approach not only caters to the needs of general users but also optimizes the expertise of professionals, maximizing their potential and value.

As a geospatial data service provider, the GDS excels in efficiently managing and visualizing data. The Geo-CMS module, a core component, processes diverse geospatial data and metadata. It includes standard WebGIS portal features like upload, map creation, and data search. The interactive mapping module integrates geospatial data with text and multimedia for an intuitive display of rich information within geospatial datasets. In addition, the data processing and analysis workflow offers a method that involves advanced data processing and publishing using local GIS software, and is then combined with visual analysis using the platform's web apps. This endeavor brings more possibilities for applying geospatial data. Another noteworthy feature is that users, leveraging the platform's social attributes, can communicate with groups, experts, and other non-specialized users to convey their ideas and requirements. This will enable the platform to gradually become a geospatial community where general users communicate and collaborate with experts to generate content.

At present, we have verified the platform's applicability in multiple scenarios. The yardangs-themed GeoStory imparts knowledge and provides users with an immersive experience by showcasing the distribution of yardangs in six typical anticline areas in the Qaidam Basin. The NDI and dashboard form a robust and flexible link between processing data and displaying information when analyzing Qi'ao Island's vegetation growth.

The GDS platform proposed in this paper explores the potential for collaboration among different users. The integration across multiple scenarios and collaboration between experts and the general public offer valuable insights for the future development of geospatial applications. In terms of future works, one promising direction is to improve support for three-dimensional spatial data, while another is to improve the management and creation of metadata. Ultimately, we endeavor to create a more open geographic community.

**Author Contributions:** Conceptualization, Ninghua Chen, Jianyu Chen, and Wenqi Gao; methodology, Wenqi Gao, Ninghua Chen, and Jianyu Chen; data curation, Wenqi Gao, Xuhua Weng, and Xinhao Jiang; software, Wenqi Gao; investigation, Wenqi Gao; validation, Wenqi Gao, Ninghua Chen, and Jianyu Chen; formal analysis, Wenqi Gao; resources, Ninghua Chen and Jianyu Chen; writing—original draft preparation, Wenqi Gao; writing—review and editing, Wenqi Gao, Ninghua Chen, Jianyu Chen, Bowen Gao, and Yaochen Xu; visualization, Wenqi Gao, Xuhua Weng, and Xinhao Jiang; supervision, Ninghua Chen; project administration, Jianyu Chen; funding acquisition, Jianyu Chen. All authors have read and agreed to the published version of the manuscript.

**Funding:** This research was supported by the NSFC-Zhejiang Joint Fund for the Integration of Industrialization and Informatization (Grant U1609202) and the National Natural Science Foundation of China (Grants 42076216, 41376184 and 40976109).

**Data Availability Statement:** Data are contained within the article.

**Acknowledgments:** We thank the editors and the anonymous reviewers for their valuable comments and suggestions.

**Conflicts of Interest:** The authors declare no conflicts of interest.

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
