# Peer review of "A Novel and Extensible Remote Sensing Collaboration Platform: Architecture Design and Prototype Implementation"

_ijgi, doi:10.3390/ijgi13030083_

Round 1

Reviewer 1 Report

Comments and Suggestions for Authors

The GDS Platform: It will be helpful if the GDS Platform website is shared within this article  to allow the reader to access it and make a direct review on it, to further understand and internalize the Figures attached in the article. The reviewer will have the opportunity to see the platform in action and try out a number of its operations and appreciate and feel its general outlook.

Referencing: On Line 38 reference is made to [5] without a name, however in Line 39 and 40 names are mentioned along Ref [6] and [7]. The same goes for Ref[8]. Consistency is viewed as more palatable to the reader.

Professional Users: the review of the letirature refers to professional users. Since professional users are many, it will be helpful to specify the professionals the writers are referring to [Ref: Line 77 -79]

Comments on the Quality of English Language

Line 34: 'deployed' instead of "deploying"

Line 367-369: Sentence starting on line 367 and ending on line 369 must be revisited as it appears incomplete and does not completely convey what was intended to be shared.

Line 395 - 396: The sentence must be revisited and rephrased.

Reviewer 2 Report

Comments and Suggestions for Authors

The paper presents a geospatial data service (GDS) architecture based on the existing GeoNode framework that links desktop GIS software and cloud-based platforms to construct a user collaboration platform to bridge the gap between non-specialized users and experts. Two use cases including geology and ecology are shown to demonstrate the capability of the proposed platform.

Strong points:

S1. The use cases are clear and easy to follow.

Weak Points

W1. The motivation is clear but stronger evidence is needed. One of the major motivations is to make it easier for the public to use RS data and enhance collaboration between experts and non-experts. As stated in Section 2.3 (Experts then review …) and in Section 3 (by expert knowledge …), experts play an important role in the proposed platform. However, how to find and motivate these experts to help general users and take real effect in the system needs to be clarified. How to balance the workload of these experts should also be considered.

W2. The difference between the proposed work and the existing GeoNode-based spatial data infrastructure shown in Table 1 should be addressed. The appearance of the public website for GeoINTA and Geoportal Lujan de Cuyo is quite similar to the GUI of the GDS presented in Figure 2. What are the key factors that make the proposal better than existing platforms? It is stated in line 324 that “traditional geospatial application architectures are complex …” but more evidence should be provided to make it more persuasive.

W3. If possible, the link for the proposal should be provided since the authors mentioned several times that the proposed GDS platform was already in use.

Reviewer 3 Report

Comments and Suggestions for Authors

The paper represents a combination of well-known software components to re-implement already implemented mechanisms found in other solutions. Authors hardly propose novel software, methodological or architectural aspects, just an obvious combination of existing software. From this perspective, the paper has no capacity for further improvement resulting in the reject decision.

The novelty is unclear.

A more detailed review follows.

What is special in the proposed architecture? Is there any new architecture proposed at all? It is not sufficient just to deploy existing software, connect with each other, and claim that a new architecture is devised. Which new data management techniques authors propose? Conclusion claims that the solution excels in “efficiently managing and visualizing data”, but no appropriate evaluation is presented.

The description of the solution is just a name-dropping. It is hard to find any evaluation in the paper on why particular components were selected, only “the mainstream ones”, not “the appropriate ones”.

There is a large problem with the review/survey part of the manuscript. The paper does not survey related work: Microsoft, Amazon, Sentinel, EarthBlox, a wealth of NASA platforms and others. No comparison of features. Some mentioned papers are not directly related to RS data: 1,3,4 and others.

The presented motivation for creating a new platform is week: “atmospheric correction, geometric correction” were the problems decades ago, currently most datasets are distributed as atmospherically and geometrically corrected.

Reviewer 4 Report

Comments and Suggestions for Authors

In this manuscript, a geospatial data service architecture is designed to construct an efficient user collaboration platform. Based on the scalability of the platform, three web apps have been developed for the different themes of ecology, oceanography, and geology respectively. In this pilot phase, the proposed platform explores the potential for collaboration among different users and hereby proves that the gap between non-specialized users and experts is successfully bridged, demonstrating the platform's powerful interactivity and visualization.

As for the general writing, the article is well structured and the main contribution of is clear. However, the quality of this manuscript has been reduced by the following two flaws.

1.       The authors have developed three Apps for the different themes of ecology, oceanography, and geology. In the section of ‘4. Use Cases’, however, there are merely two cases presented - ‘4.1 Geology case and 4.2 Ecology case. There is no use case concerning on oceanography, why?

2.       The authors just mentioned “Currently, we apply the platform to our daily work” in the part of ‘conclusion’. However, there is no concrete descriptions concerning it in the main text of the manuscript.

Given the issues mentioned above, I recommend a ‘major review’ as my general evaluation.

Round 2

Reviewer 2 Report

Comments and Suggestions for Authors

The revised version covered all my concerns. Hopefully, the link to the proposed work will be public soon.

Comments on the Quality of English Language

No further comments.

Author Response

Thank you very much for your time involved in reviewing the manuscript. We will try to add new explorations in future work.

Reviewer 3 Report

Comments and Suggestions for Authors

The revised manuscript does not address the reviewer concerns (see the previous review). The manuscript is fundamentally a description of a “yet another app”.

In the Author Response: “The idea we proposed is not just a simple combination of software but integrated innovation. This process incorporates some of the components developed by ourselves”

This does not mean that the components are scientifically or technically novel; many companies and individuals develop “new“ software based on well-known patterns and techniques.

In the Author Response: “We proposed a collaborative model”

What is the novelty in this model? Why do you call this a model?

In the Author Response: “However, most of them are not open-source solutions and cannot be used directly by small and medium-sized enterprises and teams”

Well, Microsoft Planetary Computer is based on open-source solutions, for others a proprietary code is not a disadvantage if the system is of high-quality, while open-source is not also always an advantage. Some platforms can be used by companies, e.g. GEE for business (enterprises).

New applications incorporated in the paper do not prove the significance of the work from the R&D perspective.

A scientific journal is not intended to present technical reports.

Reviewer 4 Report

Comments and Suggestions for Authors

The manuscript has been well refined. There is no further comment.

Author Response

(The authors gave the same response as above.)
